# Sustainability of Cocoa (*Theobroma cacao*) Cultivation in the Mining District of Ponce Enríquez: A Trace Metal Approach

**DOI:** 10.3390/ijerph192114369

**Published:** 2022-11-03

**Authors:** Carolina Ramos, Jeny Ruales, José Luis Rivera-Parra, Masayuki Sakakibara, Ximena Díaz

**Affiliations:** 1Área de Ambiente y Sustentabilidad, Universidad Andina Simón Bolivar, Toledo N22-80 (Plaza Brasilia), Quito 170525, Ecuador; 2Departamento de Ciencia de Alimentos y Biotecnología DECAB, Escuela Politécnica Nacional, Ladrón de Guevara E11-253, Quito 170525, Ecuador; 3Departamento de Petróleos, Escuela Politécnica Nacional, Ladrón de Guevara E11-253, Quito 170525, Ecuador; 4Department of Earth Science, Graduate School of Science and Engineering, Ehime University, Matsuyama 790-8577, Japan; 5Faculty of Collaborative Regional Innovation, Ehime University, Matsuyama 790-8577, Japan; 6Research Institute for Humanity and Nature, Kyoto 603-8047, Japan; 7Departamento de Metalurgia Extractiva, Escuela Politécnica Nacional, Ladrón de Guevara E11-253, Quito 170525, Ecuador

**Keywords:** cocoa, contaminant mobility, contamination, health risk, toxic metals

## Abstract

Historically, cocoa (*Theobroma cacao*) has been one of Ecuador’s most important export crops. In the Ponce Enriquez district, artisanal and small gold mining (ASGM), and quarrying account for 42% of economic activities, while agriculture and livestock farming account for 30%, making the analysis of their synergy and interaction key to understanding the long term viability of the different activities. In this study, we evaluated the concentration of potentially toxic metals in different parts of the cocoa plant and fruit, in relation to mining activities within the area. Gold extraction generates pollution, including potentially toxic metals such as mercury (Hg), cadmium (Cd), arsenic (As), copper (Cu), lead (Pb) and zinc (Zn). In order to understand the mobility of these metals within the cocoa plant and fruit, the analysis was conducted separately for leaves, pod, husk and cocoa bean. Concentrations of the target metals in the different plant parts and soil were measured using ICP-MS, and the mobility and risk factors were calculated using the transfer factor (TF) and the risk ratio (HQ). The results suggest that Zn, Cd and Cu are indeed moving from the soil to cocoa leaves and beans. Furthermore, the results show that the concentrations of toxic metals in the different parts of the cocoa fruit and plant, particularly in the cocoa bean, which is used for chocolate manufacture, are not higher than those regulated by FAO food standards, as is the case of Cd, which is limited to 0.2 mg Cd/kg and in the samples analyzed does not exceed this limit. Even though the concentration of these metals does not exceed the safety standard, the presence of these potentially hazardous metals, and the fact they are absorbed by this important local crop, are worrying for the long-term sustainability of cocoa cultivation in the area. Therefore, it is fundamental to monitor the local environment, understanding the distribution of heavy metal pollution, and work with the local authorities in landscape management to minimize the exposure of crops to ASGM pollution.

## 1. Introduction

Cocoa (*Theobroma cacao*), an ancestral crop widespread around the American tropical countries, presents a historical and cultural richness. Initial references date from the first years after the arrival of the Spaniards, hence its importance for farmers and for trade [1].

Ecuador is a cocoa producing country with more than 240 years in the international market. Cocoa beans export significantly and have contributed to the country’s economy, especially in its first and second “Cocoa Boom” [2]. It is currently the fourth cocoa producing country in the world, with 300,000 tons per year [3]. 

The environmental impact of gold extraction generated by artisanal and small gold mining (ASGM) is a global concern. Thus, several countries are trying to mitigate the environmental and health problems that arise, particularly if mercury amalgamation is used for gold purification [4]. In Ecuador, gold mining has existed since pre-Columbian times [5] and is currently carried out in a small scale, which varies between artisanal mining related to illegal practices to small scale formal mining [6]. The migration process of artisanal miners to the Ponce Enrique mining district was a boom in the 1980s, which triggered a concern for its potential detrimental effects and motivated a search for responsible and environmentally sustainable activities [5].

Since mining is seen as an activity that generates significant economic income for a community, preventing its impact on water and soil quality, biodiversity, and the basis of human life, should be a priority [7]. The main resource affected is water, because mining modifies the composition and quality of local rivers, causing an increase of dissolved solids, metalloids and cations concentrations rendering it unsafe for human use, especially since these same water sources are used for irrigation and human consumption [8,9].

This study focuses on one of the main crops within the study area, cocoa. In order to evaluate the sustainability of cocoa cultivation in the area, we started with the identification of environmental and anthropic factors that affect production, such as land use, the availability of irrigation water, and other economic activities that are carried out in the area [10]. One of those activities is artisanal and small-scale metallic mining, which generates significant environmental impacts [9,11]. Moreover, there is evidence that gold mining in the Ponce Enriquez county has led to the contamination of the surface water systems in the area [9,12], generating problems in the quality of the water. Use of this water becomes potentially dangerous for human and environmental health, generating concerns for the sustainability of cocoa and other crops production.

The permanent demand for food, water and ecosystem services induces impacts on the structure, distribution and functioning of resources, generating constant pressure due to human activities, causing a deterioration of the ecosystem, and jeopardizing the availability of food [13]. Therefore, considering a major local crop and the potential presence of contaminants is an interesting approach for understanding its long term sustainability, and to start understanding the sustainability of the local economic fabric.

There are international standards, such as CODEX STAN 193-1995, that regulate the content of metals in cocoa, such as lead and cadmium. Moreover, the European Union standard regulation N° 488/2014 came into force on 1 January 2019, which includes limits for metal concentrations in chocolate, one of cocoa’s main products. [14]. Therefore, metal concentration analysis became important for cocoa exports. Furthermore, there is evidence that other metals such as mercury or magnesium, which are known to cause detrimental health effects, can be absorbed by the cocoa plant and end up in its products, consequently with potential health risks [15].

In general, for an analysis of major and trace elements of natural or anthropogenic origin, those classified as toxic, which are related to potential health risks, such as Hg related to Minamata syndrome, and As related to skin keratosis, etc., will be used [16].

In studies in Indonesia, the concentration of Hg in soil in the mining area is classified as critical (Hg > 0.3 ppm), as well as in plant tissue (Hg > 0.03 ppm). Strategic regulation is therefore recommended to protect people from mercury exposure, to maintain agricultural activity and to minimize environmental damage and threats from mining production [17].

This study aims to assess the sustainability of cocoa (*Theobroma cacao*) cultivation in the mining district of Ponce Enriquez. Understanding the sustainability of cocoa cultivation is essential when analyzing economic activities in Ponce Enriquez because two activities, mining and agricultural production (cocoa and banana), are the basis of the local economy and contribute to the development of the community.

## 2. Materials and Methods

### 2.1. Study Area

The present research was carried out in the Azuay province, in the mining district of Camilo Ponce Enriquez. The simple random model was used for sampling distribution. Eighteen productive cocoa farms were identified and sampled (Figure 1). The identified main water basin that would influence cocoa crops in the sampled areas is the Gala River. The Chico River and Siete River, which are the main tributaries to the main basin, are directly affected by ASGM activities.

The study area has a tropical climate. The average annual temperature is 27° and the average annual rainfall is 364 mm, the average humidity is 80% and the UV Index is 6. There are two main seasons, rainy (October to April) and dry (May to September) [12].

### 2.2. Field Sampling

The cocoa farms are scattered throughout the Camilo Ponce Enriquez mining district. Sampling was conducted in the months of March and July 2019. Each sampling site was identified and georeferenced in situ. Samples were taken from 18 farms identified in a random sampling, as indicated in Figure 2. Soil samples were taken near the sampled plant at 20 cm and 50 cm depth with a drill. These samples were stored in resealable bags and kept refrigerated (4 °C). The fruit samples were further separated into three subsamples: almonds (defined as the cocoa bean without the husk), shell and husk. These samples were frozen at −80 °C, together with leaves from each sampled plant.

### 2.3. Laboratpry Analysis

All the samples were freeze-dried (LEYBOLD freeze drier, model ELEKTROVER GT2) in a process lasting 72 h. The dried samples were crushed in an agate mortar, which was previously and thoroughly cleansed to avoid cross contamination. The crushed and homogenized material was used for subsequent acid digestion in a microwave oven (MILESTONE ETHOS UP) using 500 mg of sample with 9 mL of nitric acid (65% *w*/*w*, Fisher Chemical, Waltham, MA, USA), and 1 mL of hydrogen peroxide. The samples were gauges to 50 mL with type I water and stored in Falcon tubes at 4 °C.

Soil samples were dried at room temperature (19 °C on average) for 4 weeks. The dried samples were pulverized and then acid digested in a microwave oven (MILESTONE ETHOS UP). Acid digestion was performed by mixing 150 mg of homogenized soil sample with 6 mL of HNO_3_ (65% *w*/*w*, Fisher Chemical) and 2 mL of hydrochloric acid HCl (37% *w*/*w*, Merck, Rahway, NJ, USA). The extract obtained was gauged to 50 mL with type I water. The extract was transferred to a falcon tube and stored under refrigerated storage at 4 °C. 

In the case of mercury analysis, it was performed by means of cold vapor atomic fluorescence spectroscopy. Major and trace metals analysis were performed in an Agilent 7500 ce ICP-MS (Agilent Technologies, Inc., Santa Clara, CA, USA). Interferences were minimized by collision or reaction with gas in a collision cell. Indium (10 ng/L equivalent concentration) was used as internal standard.

Quality control was carried out using the EPA Contract Laboratory Program Statement of Work for Superfund Analytical Methods (Multi-Media, Multi-Concentration), SFAM01.0 Exhibit D Inorganic Methods, Inductively Coupled Plasma—Mass Spectrometry Metals Analysis, released in May 2019.

The samples used for QA/QC (quality assurance/quality control) included an initial calibration blank (ICB), initial calibration verification (ICV), CRQL check standard (CRI), continuing calibration verification (CCV), continuing calibration blank (CCB) and interference check sample (ICS). For each 10 samples, a duplicate, spike, spike duplicated, serial dilution, CCV and CCB were run.

### 2.4. Data Analysis

All statistical tests were conducted using IBM SPSS Statistics 25.0 Core System, EEUU. The SPSS (Statistical Package for the Social Sciences) software was used for statistical analysis on the concentrations for each sample (soil at 20 cm, soil at 50 cm, leaves, husks, shells and cocoa beans). In order to evaluate if there is a significant difference between the soil samples, an analysis of the means was performed. Metal mobility through the plant and fruit was determined by linear correlation through the different plant parts along a metabolic gradient, going from the leaves to the most internal parts of the fruit.

For the mobility analysis, the transfer factor (TF) was used. Equation (1) describes TF as the ratio between metal concentration in the cocoa beans or leaves and the concentration obtained in the soil sample. If TF is greater than 1, it means that the metal has the capacity to move from the soil to the aerial part of the plant being analyzed [18]:(1)$$\mathrm{TF}=\frac{C_{\mathrm{plant}\;}\;\left(\frac{\mathrm{mg}}{\mathrm{kg}}\right)}{C_{\mathrm{soil}}\;\left(\frac{\mathrm{mg}}{\mathrm{kg}}\right)}$$
where

C_plant_ = Metal concentration in different parts of the plant.

C_soil_ = Concentration of metal in the soil.

In order to estimate the risk to human health the chronic daily intake index (CDI) was calculated using Equation (2) [17]:(2)$$\mathrm{CDI}=C\left(\frac{µg}{L}\right)*\frac{\mathrm{DI}\;\left(\frac{L}{\mathrm{day}}\right)}{\mathrm{BW}\;\left(\mathrm{kg}\right)}$$
where

C = Concentration of the metal of interest, 

DI = Average daily intake

BW = Average body weight

To calculate the chronic daily intake (CDI), the average body weight reported by [19] are used (70 kg for adults and 15 kg for children). Similarly, the average daily intake used is 2 L/day for adults and 1 L/day for children. The units of measurement are expressed in L considering that the density of the liquid resulting from acid digestion is 1 kg/L.

The risk ratio HQ (Equation (3)) is calculated, where RfD represents the reference dose of toxicity of an element. When calculating the HQ value, if this value is less than 1, the population is healthy and safe, regarding metals such as Cd, Pb, Cr and Fe [17]. This ratio is relevant as it helps us to identify metals that potentially become health hazards:(3)$$\mathrm{HQ}=\frac{\mathrm{CDI}\left(\frac{µg}{\mathrm{kg}*\mathrm{day}}\right)}{\mathrm{RfD}\left(\mathrm{mg}/\left(\mathrm{kg}*\mathrm{day}\right)\right)}$$
where

CDI = Chronic daily intake

RfD = Toxicity reference dose of an element

## 3. Results

### 3.1. Metal Content

Our analysis focused on Hg, Cd, As, Cu, Pb and Zn. These elements were chosen considering they have been determined as contaminants generated from mining activity and classified as potentially toxic [16]. Figure 3 plots these metals in a range of 0 to 1.2 ppm concentration. 

Metals such as Hg and Cd present greater content in cocoa beans than in the husk, leaves and shell, generating concerns for their presence in processed products such as chocolate (Figure 3).

In the case of Cu, concentration values are maintained throughout the crop system (Figure 3). Although its concentration is not higher than the safety standards, it should be monitored regularly because at high concentrations in cocoa beans it can cause different alterations in plants and human health, such as a decrease in growth rate, hypochromic anemia and diarrhea, among others [20].

Zn is a micronutrient needed for plant growth that directly affects the quality of cocoa beans [21]. Its distribution in the plant components is not uniform, with higher concentrations in leaves and lower concentrations in surface soils (Figure 3).

The concentration of these metals (Zn, Cu, Hg, As, Pb and Cd) in the leaves of the crop highlights the importance of management practices. Regular pruning is practiced during agricultural work, and this plant material is deposited around the plant itself. Thus, the absorbed metals may be incorporated back into the soil and the plant itself.

### 3.2. Mobility Analysis

To analyze the mobility of metals within the plant, the transfer factor (TF) is calculated. The transfer factor varies from 0 to approximately 3.5 between cocoa beans and the soil (Figure 4). Metals presenting mobility capacity are mainly Cd, Zn and Cu. Hg showed high mobility in only one sample.

It is important to analyze mobility between the soil and the cocoa beans because the soil is consumed and becomes the medium for the metals to enter the organism, and during the production cycle the cocoa beans remain on the plant for a short time, which indicates how quickly the metals are mobilized in the plant.

Regarding the transfer factor calculated from the concentration of the leaves and the soil, Figure 5 shows that mainly Zn, Cd and Cu could move from the soil to the leaves. These values of the transfer of metals to the leaves are used as an alternative for a bioremediation proposal. Considering that metals accumulate in leaves, this plant material can be pruned and removed to prevent the transferred metals from returning to the soil, as Pb and Hg seem to be mobilized in few of the samples from the soil to the leaves.

The maximum permitted concentration limits of metals for plant material, according to the FAO, establish standards for Cd (0.2 mg/kg) and Zn (99.4 mg/kg) [22].

With regards to Cd (Figure 6), the concentration in the culture system is below the permitted level (0.2 mg Cd/kg) [22] for the samples analyzed. Although the concentrations are below the permitted level, they should be monitored to prevent anthropogenic conditions from altering these levels and making the concentration toxic to humans. The observed higher Cd concentration in cocoa beans than in the soil, as well as in the case of the leaves, allow us to conclude that Cd is being bioaccumulated during its productive cycle in these parts of the plant.

The concentration of Zn in the crop system (Figure 7) is below the permitted level for this metal (99.4 mg Zn/kg) [22]. It is observed that for some cases the concentration in the cocoa beans is higher than in the soil, as in the case of the leaves, concluding that it is being bioaccumulated during its productive cycle in these parts of the plant.

However, previous studies show that Zn is an element that moves little in the plant but has many important functions. The structure and functionality of many enzymes depends on the presence of Zn in plants, so the concentration present in the plant does not affect it [23].

### 3.3. Risk Analysis

Table 1 presents the data obtained from the risk analysis of the calculated metals of interest. Based on the risk ratio for the non-cancer risk, HQ (hazard quotient), the oral toxicity reference dose, RfD, according to the USEPA were considered [17] for the different metals. Results show HQ values less than one in all cases, which indicate that the exposed population can be considered safe from a non-carcinogenic risk, when ingesting the metal concentration found in the cocoa beans. For the calculation of chronic daily intake (CDI); the average body weight reported by [19] for adults is 70 kg and for children 15 kg is used. For the case of average daily intake was used for adults 2 L/day and for children 1 L/day.

## 4. Discussion

The metal concentrations analyzed were not higher than those established in food safety standards. However, frequent monitoring should be carried out to ensure that those concentrations do not increase and become a health problem. Specifically, the concentration of toxic metals in cocoa beans, although they do not exceed the permitted toxic levels, should be monitored, mainly because during the transformation process to chocolate, they might increase. This is similar to what happens in the case of Pb in rice, where levels increased during processing [24]. Additionally, the concentration in all waste material generated in cocoa processing should be monitored, specifically in cocoa husks, because they are used as raw material for packaging, animal feed and in some cases as an alternative source of fiber for food [25]. So, they may become a potential hazard for consumers if the toxic metal concentrations exceed the permitted levels after transformation.

Considering that Cd is regulated for chocolate, the main processed product of cocoa, it is necessary to generate alternatives to reduce the absorption of this metal by the plant. A common agricultural task is pruning; this activity consists of removing plant material and placing it as soil cover near the plant, helping the crop due to the flow of mineral nutrients produced in the biogeochemical cycles present in natural or agroforestry systems and reducing evaporation [26]. As a natural phytoremediation mechanism, it is recommended to remove the leaves when pruning and not to leave them on the soil, to avoid Cd and other metals being released back to the substrate where the plants are growing. This removed plant material should be treated, considering its concentration of heavy metal concentration. An alternative within our study area is to deposit the material in the tailing ponds managed by ASGM projects, where it could go through composting processes as a pretreatment, then be incinerated, which will significantly reduce the contaminated biomass [27]. However, a balance must be obtained between removing toxic metals present in the leaf litter and taking advantage of the nutrients. Furthermore, phytoremediation processes must be continually evaluated because they may suffer interference in their efficacy due to the appearance of new contaminants from emerging technologies containing trace metals [28].

The presence of toxic metals in soil, water and air has different origins, the natural ones need to be monitored to assess their influence on the environment; but those of anthropogenic origin such as those generated in industries, mining activities, agriculture, etc., generate an urgent need to reduce the amount, before it is too late. Therefore, sustainability measures should focus on reducing the use of fertilizers and pesticides [29].

The analysis of the composition and concentration of toxic metals of natural origin depends on the type of rock, as well as the environmental conditions that cause weathering processes. Metals from the geological environment present in the plant matter can be Cd, Pb, Co, Mn, Cr, Zn, Cu, Ni, Hg and Sn [30]. Therefore, it is suggested to evaluate the origin of Cd, Pb and other metals, that are observed to be mobilized in the plant. Metal content in water in highly mineralized areas, such as the Camilo Ponce Enriquez mining district, are expected to be naturally high, because of natural geochemistry and the weathering process [31]. Therefore, the high metal content in the soil and corps may also be affected by water irrigation.

The major concern regarding the presence of heavy metal pollution in the different environmental compartments in the area is worrisome, from a public health perspective, due to a potential bioaccumulation and long-term toxicity. Orchard plants, such as spinach, tomatoes and peppers are prone to incorporate toxic metals into their tissues, becoming a risk for local farmers [31].

Even though toxic metals and metalloids are silently present in all environmental compartments, their detrimental effects can be felt through generations and even be exported as part of a crop like cocoa beans. Monitoring and controlling the environmental problems, in areas such as Camilo Ponce Enriquez, could make it possible to speak of a coexistence between agriculture and mining [32]. High impact activities such as mining are keystones of our modern way of life. Therefore, understanding its effects on complex land use matrixes, such as our study area, opens the possibility to test the management and technological alternatives to achieve long term sustainability, which would support the local economy and the community well-being.

## 5. Conclusions

The analysis of toxic metals is important because of their bioaccumulation. Bioaccumulation implies an increase in the concentration of an organism over time, relative to the concentration in the environment.

The concentration of the metals analyzed were not higher than those mentioned in food standards, however, frequent monitoring should be carried out to prevent the metal presence becoming a health problem.

Of the metals analyzed, Cd was found to be mobile within the plant. As part of the agricultural work, it is recommended to remove the leaves when pruning and not leave them on the ground, as a natural phytoremediation mechanism.

Caring for the environment is important for the future of mankind, but it is crucial that remediation measures are continually being evaluated to ensure they work effectively and help to care for natural resources. 

The occurrence of toxic metals in soil, water and air has different origins; the natural ones need to be evaluated on the influence on the environment and those of anthropogenic origin create an urgent need to reduce them before it is too late to achieve crop sustainability.

From the mobility and risk factor values, it is observed that Zn, Cd and Cu move from the soil to leaves and cocoa beans. In addition, the results show that there are concentrations of toxic metals in different parts of the cocoa fruit and plant, particularly in the cocoa bean. This generates concern for the cocoa processing industry, because their products (chocolate) could have concentrations higher than those regulated by FAO food standards, due to bioaccumulation.

Soil concentrations and external conditions (water source, suspended solids in the air) should be monitored to try to determine the origin of increased concentrations of these metals in cocoa beans, compared to the soil.

## Figures and Tables

**Figure 1 ijerph-19-14369-f001:**
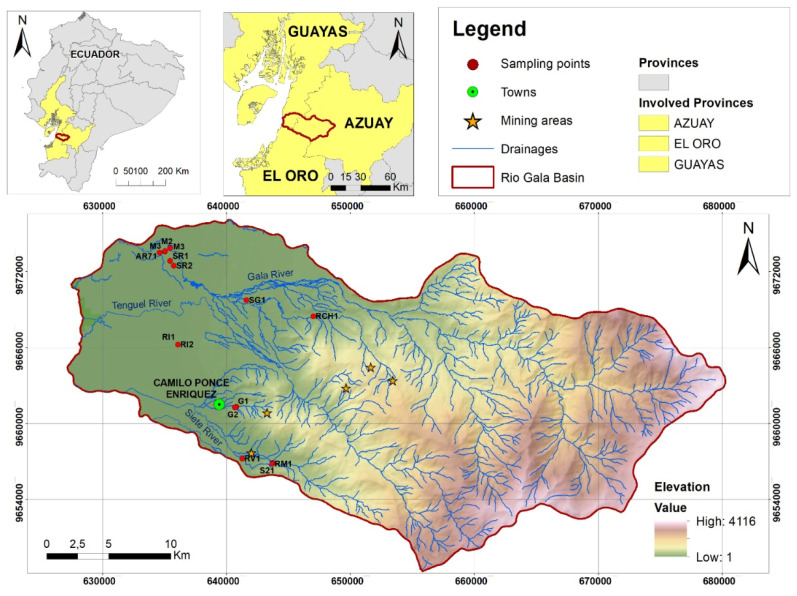
Macro location of sampling points. Mines near the sampling site. Sampling sites.

**Figure 2 ijerph-19-14369-f002:**
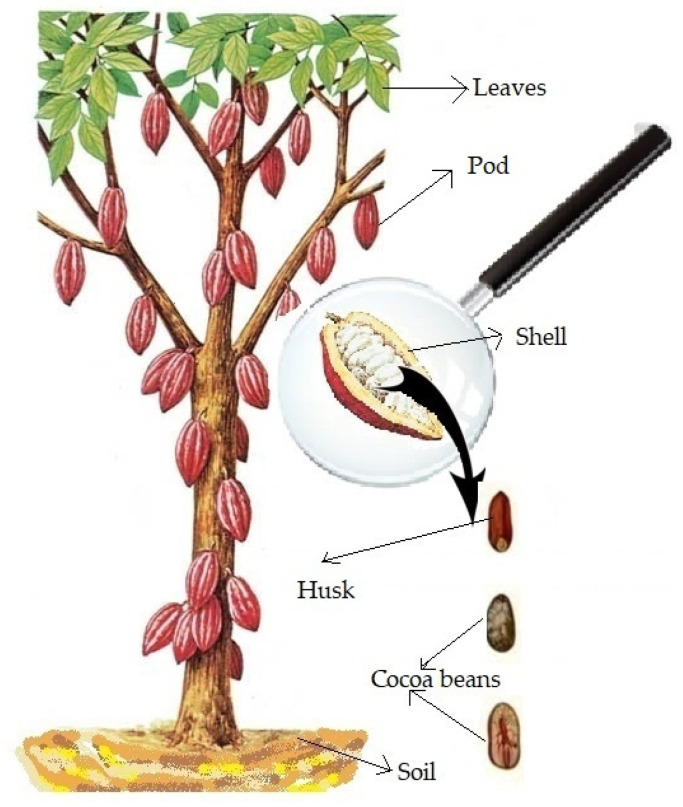
Parts of cocoa plant sampled.

**Figure 3 ijerph-19-14369-f003:**
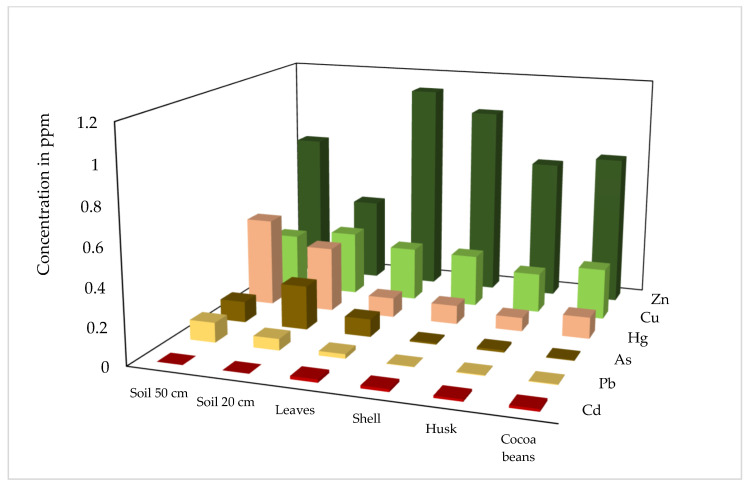
Metal content in soil and different parts of the cocoa plant.

**Figure 4 ijerph-19-14369-f004:**
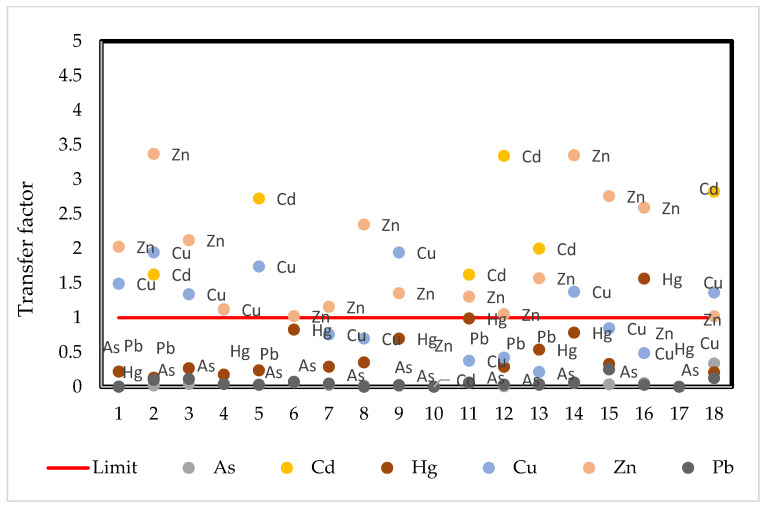
Transfer factor for cocoa beans-soil. The red line represents the limit above which metals are observed to be able to be transferred in the plant.

**Figure 5 ijerph-19-14369-f005:**
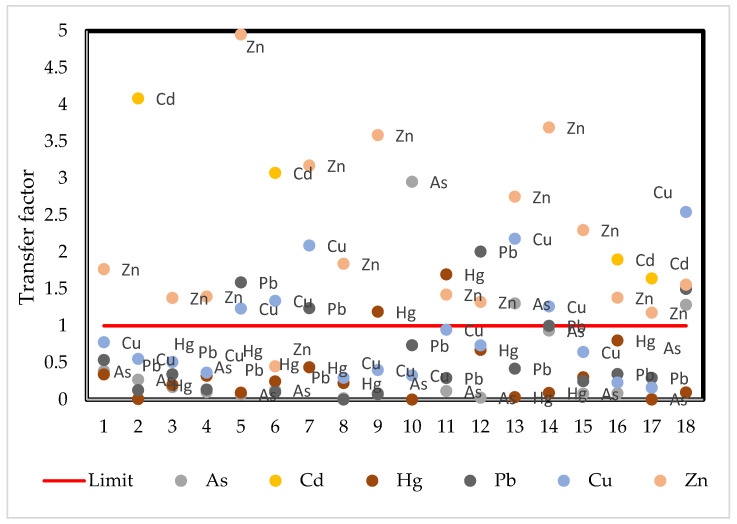
Transfer factor for leaves-soil. The red line represents the limit above which metals are observed to be able to be transferred in the plant.

**Figure 6 ijerph-19-14369-f006:**
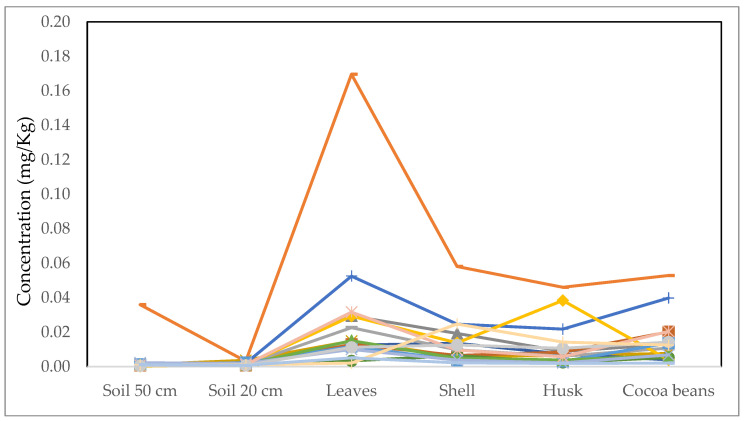
Cd content in soil and different parts of cocoa plant, the concentration is observed for 18 samples. The limit is 0.2 mg/kg, and no sample exceeds this level.

**Figure 7 ijerph-19-14369-f007:**
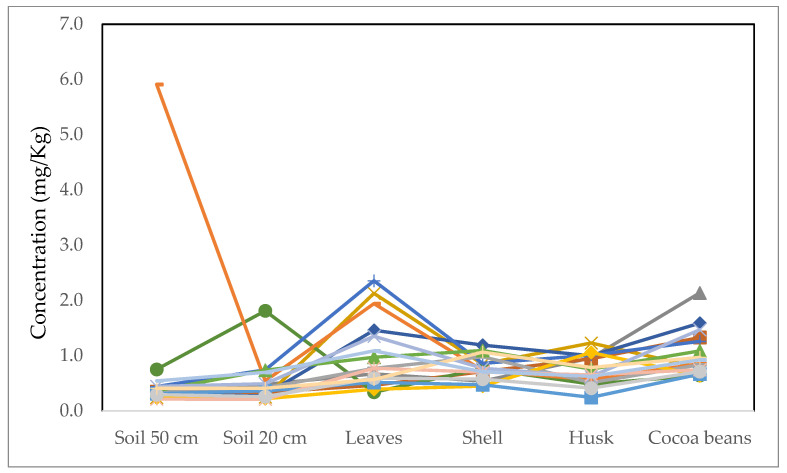
Zn content in soil and different parts of cocoa plant, the concentration is observed for 18 samples. The limit is 99.4 mg/kg, and no sample exceeds this level.

**Table 1 ijerph-19-14369-t001:** Hazard quotients and total hazard index for heavy metals.

	Cd	Pb
$\bar{C}$ (µg/L)	1.31 × 10^−5^	3.56 × 10^−6^
IDU Adults	1.07 × 10^−5^	2.91 × 10^−6^
CDI Children	5.84 × 10^−5^	1.58 × 10^−5^
RfD (mg/(kg-day))	5 × 10^−4^	3.5 × 10^−3^
HQ Adults	2.14 × 10^−2^	8 × 10^−4^
HQ kids	11.67 × 10^−2^	4.5 × 10^−3^

## Data Availability

Even though the data is not publicly available it can be requested to the corresponding author.

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
