# Peer review of "Sustainability of Cocoa (Theobroma cacao) Cultivation in the Mining District of Ponce Enríquez: A Trace Metal Approach"

_ijerph, 2022, doi:10.3390/ijerph192114369_

Round 1
Reviewer 1 Report (Previous Reviewer 3)
The manuscript (MS) “Sustainability of cocoa (Theobroma cacao) cultivation in the mining district of Ponce Enríquez: A trace metal approacht” evaluated the concentration of toxic metals in the cocoa cultivation in the mining district of Camilo Ponce Enriquez. The topic fits the aims and scope of the IJERPH.
In my point of view, the authors of MS tried to improve the article, but this is not enough. MS still needs in some transformation.
Here is a list of corrections.
Title: "A trace metal approach". I recommend to change title, because MS deals with several trace metals but not onle one. Also it would be usefull to indicate in title what parts of cacao trees were studied (beans, leaves...) and soils (or topsoils).
Abstract: Please, add more main results: origin of TEs, comparison of concentration of TEs in beans and soils with the Ecuadorian limits, correlations with TEs concentration with each others and in plants and soils.
Introduction: Please add information about previouse studies of cacao in Ecuador. The authors paid particular attention to mercury, but cocoa beans may also contain toxic cadmium and other elements that have been studied. It is necessary to show a reason of the TEs choice and the relevance of their study (human toxicity and ecotoxicity).
Methods. Map should be corrected. It is needed to show sampling sites, the location of gold mine and nearest settlements. Where was located a control area?
Please, add detail information about each sampling - soils, parts of cacao tree and others. Add information about Limits of Detection (LD), Quantification (LQ), certified and measured concentrations (mean ± SD) of trace elements in certified standards (SRM or anothers, pls, indicate) analyzed by ICP-MS.
Please, add more information about Hg analysis methods. Did you use fluorescence spectroscopy?
What cacao varieties were collected?
Results and discussion: Did you compared TEs contents in cocoa with the CODEX Alimentarius guidelines, and in soils - with background?
What were the the origins of each TEs? The potential sources of TEs could be a combination of parent rock material, volcanic emissions, oil activities, agrochemical products, among others. It would be useful for explaining TEs spatial variability.
The uptake capacity of TEs varies between genotypes of the same plant species ( Barraza, F., Schreck, E., Uzu, G., Lévêque, T., Zouiten, C., Boidot, M., Maurice, L.Beyond cadmium accumulation: Distribution of other trace elements in soils and cacao beans in Ecuador (2021) Environmental Research, 192, art. no. 110241). Significant differences between genotypes have been found in cacao with regard to physiological traits such as leaf chlorophyll content (A.J. Daymond, P. Hadley The effects of temperature and light integral on early vegetative growth and chlorophyll fluorescence of four contrasting genotypes of cacao (Theobroma cacao) Ann. Appl. Biol., 145 (2004), pp. 257-262, 10.1111/j.1744-7348.2004.tb00381.x), stomatal conductance (A.J. Daymond, P.J. Tricker, P. Hadley Genotypic variation in photosynthesis in cacao is correlated with stomatal conductance and leaf nitrogen Biol. Plantarum, 55 (2011), pp. 99-104, 10.1007/s10535-011-0013-y). I recommend that you refer to these links and consider the results in terms of previously obtained data.
Author Response
Please see the attachment.

Reviewer 2 Report (New Reviewer)
In this study, the relationship between the concentration of potentially toxic metals in different parts of cocoa plants and fruits and mining activities in the area was evaluated. The research structure is complete, the content is substantial, and the chart information is reliable, which has certain application value in social production. However, there are still some deficiencies as follows:
1. Abstract:“Concentrations of the target metals in the different plant parts and soil were measured using ICP-MS, and the mobility and risk factors were calculated using (1) and (3) equations.”Formula (1) and (3) are inaccurate, please modify.
2. Figure 2. Macro location of sampling points:Some characters in the figure are not clear. It is recommended to update.
3. Please carefully check the paragraphs and symbols of the manuscript according to the specified format, for example, the line height of the title 2.2 does not match.
4. Please pay attention to the layout of the manuscript. For example, there is a lot of blank space on the third page.
5. The analysis and discussion in Figures 4 to 7 are few, and it is suggested to supplement the discussion on the results.
6. Is it better to mark what elements each color represents in Figure 6 and Figure 7? Suggestions for improvement.
7. 1. Introduction:There are few references in the introduction, and the introduction is generally lack of support, so it is recommended to supplement it.
8. The conclusion part is a concise summary of the full text, and it is suggested to improve the conclusion, which is divided into several points for summary.
Author Response
Please see the attachment.

Reviewer 3 Report (New Reviewer)
This study aims to evaluate the sustainability of cocoa (Theobroma cacao) cultivation in the mining district of Ponce Enriquez - Ecuador.
The manuscript describes the concentration of mercury (Hg), cadmium (Cd), arsenic (As), copper (Cu), lead (Pb) and zinc (Zn) in the cultivation of cocoa (Theobroma cacao) in the mining district of Ponce Enríquez - Ecuador. The work relates parameters such as transfer factor (TF), human health risk estimate, chronic daily intake index (CDI) and HQ risk index, in order to assess the sustainability of this crop.
The manuscript consists of abstract/keywords, introduction, materials and methods, results, discussion, conclusions, acknowledgments and references.
The text size is adequate. The language is clear. The abstract is adequate. The numbers are adequate.
The manuscript can be published in IJERPH.
However, there are some doubts about the validation of the analysis methodology:
Specificity/Selectivity, Response Function (analytical graph), Linearity, Sensitivity, Accuracy, Precision (repeatability, intermediate precision and reproducibility), Limit of Detection (LD), Limit of Quantification (LQ) and Robustness.
It is suggested that the authors mention these data in the experimental part.
Another suggestion is that in the summary of line 29 the authors replace the reference to equations 1 and 2 with the respective parameters they represent.
Author Response
Please see the attachment.

This manuscript is a resubmission of an earlier submission. The following is a list of the peer review reports and author responses from that submission.
Round 1
Reviewer 1 Report
Sustainability of cocoa (Theobroma cacao) cultivation in the mining district of Ponce Enríquez: A trace metal approach by Ramos et al. is an interesting topic to be studied. However, the manuscript is poorly written.
1. Abstract: line 26-27 re-write. confusing. Pls include human health risk values.
2. Introduction: Avoid short paras for example, line 41-43 is one line. One line can not be a para. Pls include the possible sources of metals in the study area based on land use pattern. please include what is already known from the area- review. You need to show clearly why this study has been conducted?
3. Methods: Please include climate condition of the area.
4. Results: Please organize your results based on the objectives. Figures are hardly comprehensible - in the present form these are like chaos. please see any similar published article how to present metal data.
5. Discussion : please add a comparative data with previous findings to compare your levels.
6. References : you should not use the page number in the citation. please correct.
Thank you
Author Response
Sustainability of cocoa (Theobroma cacao) cultivation in the mining district of Ponce Enríquez: A trace metal approach by Ramos et al. is an interesting topic to be studied. However, the manuscript is poorly written.
- Abstract: line 26-27 re-write. confusing. Pls include human health risk values.
The wording has been changed so that the idea is not confusing.
“regulated by the FAO food standards. Moreover, the results suggest Zn, Cd and Cu can move from the soil to the leaves and cocoa beans. Therefore, for the future of cocoa growth and sustainability, is caring and monitoring the environment”
“regulated by FAO food standards. In addition, the results suggest that Zn, Cd and Cu can move from soil to cocoa leaves and beans. For the future growth and sustainability of cocoa, it is necessary”
- Introduction: Avoid short paras for example, line 41-43 is one line. One line can not be a para. Pls include the possible sources of metals in the study area based on land use pattern. please include what is already known from the area- review. You need to show clearly why this study has been conducted?
The idea was merged to avoid short paragraphs in lines 41-42 and in lines 45-47 added about why the analysis is important in the area.
- Methods: Please include climate condition of the area.
In lines 88 to 90 you can find the climatic conditions.
- Results: Please organize your results based on the objectives. Figures are hardly comprehensible - in the present form these are like chaos. please see any similar published article how to present metal data.
The graphs presented are a summary of the data obtained, having metal concentrations with such wide ranges could not be presented in another graph, because in the proposed options for the widest concentrations are highlighted depending on the culture part; the transfer factor graph shows the amount of samples over the limit being practical in the proposed option.
It was compared with the standard levels because they are universal, studies of the area were not used because the quantification method is different from the one proposed in my research, my study is based on a sweep of metals analyzed by ICP-Ms while in the studies performed in the area are concentrated on a specific metal using AAS.
- Discussion : please add a comparative data with previous findings to compare your levels.
- References : you should not use the page number in the citation. please correct.
The citation was revised and the page number was removed.
Thank you

Reviewer 2 Report
Authors must clarified research hypothesis in the abstract
in addition research innovation potential must be explained in details in the introduction as well as in the discussion section
L52-59 authors must re-write the paragraph and to point out the impact on sustainability issues
L60-62 provide to limit information on the issue of food production
L 64 "There are international regulations for metals content..." which are those and what they mentioned ?
Material and Methods
The paper lacks clear methodology on how this study aims to assess the sustainability of cocoa. THIS IS VITAL and must explained in details . (The sustainability was asses through SPSS and analysis? this is totally inconplete)
Results section
How risk analysis were delivered ? what is the practical meaning
in addition why heavy metals play fundamental role on sustainability level. please explain.
Author Response
Authors must clarified research hypothesis in the abstract in addition research innovation potential must be explained in details in the introduction as well as in the discussion section
L52-59 authors must re-write the paragraph and to point out the impact on sustainability issues
Not rewritten, but linking the concepts presented and their relationship to sustainability as seen in lines 61-63.
“With the identification of the economic activities and environmental factors affected, the danger of sustainability can be analyzed with emphasis on cultivated agricultural products as the fundamental resources for their development are altered”.
L60-62 provide to limit information on the issue of food production
The wording was changed with a focus on food.
“In addition to the permanent demand for food, water, and ecosystem services, the impacts produced in the structure, distribution and functioning of these resources, generated by the constant pressure due to human activities, cause an ecosystem deterioration”
“On the other hand, the permanent demand for food, water and ecosystem services induces impacts on the structure, distribution and functioning of resources, generating constant pressure due to human activities and causing a deterioration of the ecosystem, jeopardizing the availability of food”
L 64 "There are international regulations for metals content..." which are those and what they mentioned ?
The suggestion was included in lines 68-71.
Material and Methods
The paper lacks clear methodology on how this study aims to assess the sustainability of cocoa. THIS IS VITAL and must explained in details . (The sustainability was asses through SPSS and analysis? this is totally inconplete)
The study makes an analysis of data provided in ICP-Ms of the concentration of metals in the crop, which allows to know how the metals are being mobilized. Sustainability is approached from a focus on the prevalence or presence of toxic metals in the crop.
Results section
How risk analysis were delivered ? what is the practical meaning
in addition why heavy metals play fundamental role on sustainability level. please explain.
The risk analysis is made according to the equation proposed in Muhammad's article, where from the concentration of metals and the constants depending on the age of the consumers, a health risk factor is obtained. A risk factor is any detectable characteristic or circumstance of a person or group of persons that is known to be associated with the probability of being especially exposed to developing or suffering from a morbid process.

Reviewer 3 Report
The manuscript (MS) “Sustainability of cocoa (Theobroma cacao) cultivation in the mining district of Ponce Enríquez: A trace metal approacht” evaluated the concentration of toxic metals in the cocoa cultivation in the mining district of Camilo Ponce Enriquez. The topic fits the aims and scope of the IJERPH.
In my point of view, MS needs in some transformation. Here is a list of corrections to be made to the text:
1. Lines 50-51 "...increase of dissolved solids, metalloids and cations concentrations..." Did authors mean cations and anions, or some special cations?
2. Calculating the CDI. Did the data on the mass of adults and children, characteristic for that region, be taken into account? The same with the norms of consumed water. I recommend to use statistics for a specific region.
3. Usually the risk is calculated from several routes of entry into the body of a toxic element: inhalation, ingestion, through the skin. Each of these paths has its own RfD = Toxicity reference dose of an element. Table with the data that was used for the calculation using equation (2) is missing. I recommend to add it.
4. I consider equations 2 and 3 are incorrect because formulas were used to calculate when drinking water containing metalloids inside, but the authors also evaluated the route of entry with cocoa beans. Water samples were not taken. If only cocoa beans were taken into account, why authors did use the norms of water consumption by the population? As a result, it seems to me, that the data turned out to be at risk is not correct.
5. Why authors did calculate non-carcinogenic risk only for lead and cadmium?
6. References 17 and 23 are the same.
Author Response
The manuscript (MS) “Sustainability of cocoa (Theobroma cacao) cultivation in the mining district of Ponce Enríquez: A trace metal approacht” evaluated the concentration of toxic metals in the cocoa cultivation in the mining district of Camilo Ponce Enriquez. The topic fits the aims and scope of the IJERPH.
In my point of view, MS needs in some transformation. Here is a list of corrections to be made to the text:
- Lines 50-51 "...increase of dissolved solids, metalloids and cations concentrations..." Did authors mean cations and anions, or some special cations?
It refers to cations and anions in general, the study did not contemplate special cations.
- Calculating the CDI. Did the data on the mass of adults and children, characteristic for that region, be taken into account? The same with the norms of consumed water. I recommend to use statistics for a specific region.
These data are not available for the region studied, so I took those recommended by the literature.
- Usually the risk is calculated from several routes of entry into the body of a toxic element: inhalation, ingestion, through the skin. Each of these paths has its own RfD = Toxicity reference dose of an element. Table with the data that was used for the calculation using equation (2) is missing. I recommend to add it.
Several inputs to the organism can be evaluated, for this study the intake of cocoa almonds was used. The data used in equation (2) is written in lines 162-164.
- I consider equations 2 and 3 are incorrect because formulas were used to calculate when drinking water containing metalloids inside, but the authors also evaluated the route of entry with cocoa beans. Water samples were not taken. If only cocoa beans were taken into account, why authors did use the norms of water consumption by the population? As a result, it seems to me, that the data turned out to be at risk is not correct.
The equations used are those suggested by Muhammad in his study, in which he is quoted as evaluating the ratio of the concentration of human consumption material and the constants. In the case of that study he uses it for water, but these equations are general.
- Why authors did calculate non-carcinogenic risk only for lead and cadmium?
Because after analysis these two metals showed mobility in the plant.
- References 17 and 23 are the same.
The suggested change was made.

Reviewer 4 Report
Manuscript Number: ijerph-1817710
[Major comments]
This manuscript is not acceptable for the publication.
Study design and contents are poor and there is no significant results and discussion. Authors should include analytical data in soil and cocoa from the reference site. Basic information, when sampling was conducted and how many samples were author analyzed, are lacking.
QA/QC is required. Especially, authors have used HCl for ICP-MS analysis and thus interference can be occurred for metal analysis (ex. As).
Bioavailable fraction of metal in soil should be used for metal transfer from soil to plant.
For health risk assessment, why the unit of concentration is “μg/L” and 2L/day or 1L/day ingestions are used? I really could not understand current approach.
Significant digit of data should be considered.
Figures 6 and 7 are not necessary. These are somehow overlapped with figure 3.
Correction of the paper by a native English speaker is strongly recommended.
Author Response
Manuscript Number: ijerph-1817710
[Major comments]
This manuscript is not acceptable for the publication.
Study design and contents are poor and there is no significant results and discussion. Authors should include analytical data in soil and cocoa from the reference site. Basic information, when sampling was conducted and how many samples were author analyzed, are lacking.
Sample data is incorporated in lines 92-94, sampling points are detailed in line 109.
QA/QC is required. Especially, authors have used HCl for ICP-MS analysis and thus interference can be occurred for metal analysis (ex. As).
These interferences were considered, but we worked with ultrapure reagents and reference materials to validate the data obtained.
Bioavailable fraction of metal in soil should be used for metal transfer from soil to plant.
Agreed, but the study did not consider the available fraction but the total concentration due to data limitation.
For health risk assessment, why the unit of concentration is “μg/L” and 2L/day or 1L/day ingestions are used? I really could not understand current approach.
With these equations one can understand the use of units in the study.
Significant digit of data should be considered.
Having data for concentrations over a very wide range it was not possible to use a standard digit number, however the suggestion was adapted to the data in Table 1.
Figures 6 and 7 are not necessary. These are somehow overlapped with figure 3.
Figure 3 expresses all metals of interest, and Figures 6 and 7 show the concentrations of two metals that show transfer capacity, so it is compared to the allowable limits.
Correction of the paper by a native English speaker is strongly recommended.
It will be sent for review when all of the above items have been corrected.

Round 2
Reviewer 1 Report
The authors have addressed almost all of the comments and significantly improved the MS
Reviewer 3 Report
The answers of the authors are unsatisfactory. The manuscript must be significantly revised and before resubmission.
My main remarks are related to the analitical methods and risk calculation. QA/QC is required. Authors should choose formulas, not just copy them.
I still do not understand why in the phrase "causing an increase in the concentration of dissolved salts, metalloids and cations" cations stand separately. lines 51-52. Dissolved salts imply the presence of cations and anions.
Manuscript should be rejected.
Reviewer 4 Report
Manuscript Number: ijerph-1817710-peer-review-v2
This manuscript is not acceptable for publication.
Authors response is not enough and not suitable.
There is no available data from the reference site.
QA/QC of ICP-MS is still not considered. No proof data in the manuscript.
Total metal in soil has been used for metal transfer from soil to plant, but authors have explained the limitation.
For health risk assessment, cocoa is powder sample, thus ingestion rate should be weight base like “μg/day” etc. But the unit of concentration is “μg/L” and 2L/day or 1L/day ingestions have been used? I still really could not understand the current approach.
Significant digit (significant figure) of data has not been still considered.
Comparison with permitted level has been shown in Figures 6 and 7. But those are not necessary. These are somehow overlapped with Figure 3 and no sample showed over the permitted level.
Which part of English has been revised?
